



# What can we learn about urban air quality
# with regard to the first outbreak of the COVID-19 pandemic?
# A case study from Central Europe

Imre SALMA[1], Máté VÖRÖSMARTY[1], András Zénó GYÖNGYÖSI[1], Wanda THÉN[2], Tamás WEIDINGER[3]

[1] Institute of Chemistry, Eötvös University, Budapest, Hungary
[2] Hevesy György Ph. D. School of Chemistry, Eötvös University, Budapest, Hungary
[3] Department of Meteorology, Eötvös University, Budapest, Hungary

Correspondence to: Imre Salma (salma@chem.elte.hu)

**Abstract.** Motor vehicle road traffic in central Budapest was reduced by approximately 50% of its ordinary level for several weeks as a consequence of various limitation measures introduced to mitigate the first outbreak of the COVID-19 pandemic in 2020. The situation was utilised to assess the real potentials of urban traffic on air quality. Concentrations of NO, $NO_2$, CO, $O_3$, $SO_2$ and particulate matter (PM) mass, which are ordinarily monitored in cities for air quality considerations, aerosol particle number size distributions, which are not rarely measured on-line continuously on longer run for research purposes and basic meteorological properties usually available were jointly evaluated. The largest changes occurred in the time interval of the severest limitations (partial lock-down in the Restriction phase from 28 March to 17 May 2020). Concentrations of NO, $NO_2$, CO, total particle number ($N_{6-1000}$) and particles with a diameter <100 nm in 2020 declined by 68, 46, 27, 24 and 28%, respectively with respect to the average reference year of 2017–2019. Their quantification was based on both relative difference and standardised anomaly. Change rates expressed as relative concentration difference due to relative reduction in traffic intensity for NO, $NO_2$, $N_{6-1000}$ and CO were 0.63, 0.57, 0.40 and 0.22 (%/%), respectively. Concentration levels of $PM_{10}$ mass, which is the most critical pollutant in many European cities including Budapest, did not seem to be largely affected by vehicles. Concentrations of $O_3$ concurrently showed an increasing tendency with lower traffic, which was explained by its non-linear reaction mechanism. Spatial gradients of NO and $NO_2$ within the city became further enhanced by reduced traffic flow, which indicates the possible role of atmospheric processes taking place in near-city background environments.



## 1 Introduction

The coronavirus disease (COVID-19) is caused by the novel, Severe Acute Respiratory Syndrome CoronaVirus 2 (SARS-CoV-2) virus. The outbreak was declared as a pandemic by the WHO on 11 March 2020 (WHO, 2020). National governments, international agencies and organisations enacted widespread emergency actions for individuals, some professionals, communities and the public to reduce the risk of infection and to combat the plague. As a consequence of the implemented measures, road traffic in many cities worldwide was reduced in a substantial manner and for a considerable time interval. In parallel, lower concentrations of several air pollutants were reported from both satellite observations and in situ measurements (Keller et al., 2020; Le et al., 2020; Lee et al. 2020; Mahato et al., 2020; Nakada and Urban, 2020; Petetin et al., 2020; Tobías et al., 2020; Wang et al., 2020).

This situation offers a unique possibility for atmospheric scientists to investigate experimentally some important atmospheric chemical and physical issues including urban air quality and climate change under extraordinary conditions of lower traffic and industrial productivity (Sussmann and Rettinger, 2020). The results and consequences of this real "ambient experiment" can be utilised to determine the true potentials of complex action plans on tranquillizing urban road circulation for handling air quality, overcrowding, traffic congestions, noise contamination and other environmental, health and climate impacts in large cities.

The task is, however, somewhat complicated. Actual concentrations of atmospheric constituents can depend on 1) their emissions from several sectors, 2) their physical removal processes, 3) local meteorological conditions mainly precipitation ($P$), wind speed (WS), planetary boundary layer height (PBLH) and atmospheric stability, 4) their (long-range) transport along trajectories and 5) possible photochemical reactions, which are largely influenced by other meteorological properties such as global solar radiation (GRad), relative humidity (RH) and air temperature ($T$), and by availability of and interactions with other chemical species present in the air. Many of the phenomena or properties listed are, in addition, interconnected and confound, which further obscures the situation since they create an internally interacting non-linear environmental system.



Tropospheric residence time of constituents can also play a role under non-steady-state
conditions (Harrison, 2018). As a result, atmospheric concentrations at a fixed site change both
periodically and randomly (fluctuate) on daily, seasonal or annual scales. The variations are
also linked to the geographical location and features of urban site (de Jesus et al., 2019).

Source-specific markers generated by internal combustion engines or added on purpose into
their fuel (e.g. Horvath et al., 1988; Gentner et al., 2017) or multivariate statistical methods
(Hopke, 2016) can be applied to estimate the importance of vehicle traffic for air quality. These
methods usually require advanced analytical methods to obtain data for specific species, which
may not be available with a required time resolution or need a larger number of data, which
can be constrained by duration of time intervals of interest. Another possibility is to evaluate
jointly the time series of existing complex atmospheric data sets. This approach (described later
in more detail) can be utilised retrospectively and it is generally applicable in different cities in
the world, which were affected by road traffic restrictions.

In Hungary, state of emergency was introduced on 11 March 2020. It involved sequential
closure of education institutes, beginning of work-from-home and social distancing. It was
followed by restrictions on movement. During this, residences could only be left with specified
basic purposes, administrative centres, restaurants and touristic places were closed, distant
travels were ceased, public parks were closed for long weekends and there were various time
limitations on shopping. The mitigating measures resulted in perceivable changes in vehicular
road traffic and atmospheric concentrations. The main objectives of the present paper are 1) to
introduce and demonstrate a general method for quantifying concentration changes, 2) to
evaluate whether the changes observed were related to motor vehicle road traffic, 3) to assess
the effect of traffic on these alterations, and 4) to estimate and debate the potentials of
tranquillized urban vehicle flow on air quality.
**2 Methods**
Criteria air pollutants, namely $NO$, $NO_2 = NO_x - NO$, $CO$, $O_3$, $SO_2$ and particulate matter (PM)
were involved in the study. The species originate from different sources. Vehicular road traffic
is associated with $NO$ and $CO$, while $NO_2$ and $O_3$ are formed by chemical reactions in the air.
Contributions of residential heating, cooking, industrial activities, regional traffic in winter and
secondary processes to $PM_{2.5}$ mass are of large importance in many cities, including Budapest.




At the same time, $PM_{10}$ mass represents disintegration sources, e.g. windblown or resuspended
soil, crustal rock, mineral and roadside dust particularly under dry weather conditions,
agricultural activities in the region and material wear (Putaud et al., 2010; Salma et al., 2020a).

Aerosol particle number concentrations in the diameter ranges from 6 to 1000 nm ($N_{6-1000}$) and
from 6 to 100 nm ($N_{6-100}$) are mainly assigned to high-temperature emission sources (such as
road traffic or incomplete burning) and atmospheric new particle formation and growth (NPF)
events (Paasonen et al., 2016; Rönkkö et al., 2017; Salma et al., 2017). The latter process occurs
as a daily phenomenon with a typical shape of its monthly occurrence frequency (Salma and
Németh, 2019). This distribution changes in Budapest from year to year without any
tendentious character (Salma et al., 2020b). Particles with a diameter from 25 to 100 nm ($N_{25-100}$
in cities are mainly emitted by incomplete combustion (or consist of grown new particles
by condensation), while the size fraction with a diameter from 100 to 1000 nm ($N_{100-1000}$)
expresses physically and chemically aged particles, which usually represent larger spatial
extent (Salma et al., 2014; Mikkonen et al., 2020).

Approximate tropospheric residence time of $NO_x$, CO, $O_3$, $SO_2$ and PM are estimated to 1–2
days, 2 months, 1–2 months, 4–12 days and from several hours up to 1 week depending largely
on particle size, chemical composition and environment, respectively (Warneck and Williams,
2012; Harrison, 2018).

## 2.1 Experimental data

The concentrations of $NO/NO_x$, CO, $O_3$, $SO_2$, $PM_{10}$ mass and $PM_{2.5}$ mass were measured by
UV fluorescence (Ysselbach 43C), chemiluminescence (Thermo 42C), IR absorption (Thermo
48i), UV absorption (Ysselbach 49C), beta-ray attenuation (two Environment MP101M
instruments with $PM_{10}$ and $PM_{2.5}$ inlets) methods, respectively with a time resolution of 1 h.
The particle number concentrations were determined by a flow-switching type differential
mobility particle sizer (DMPS; Salma et al., 2016b) with a time resolution of 8 min. The latter
measurements were performed in a diameter range from 6 to 1000 nm in 30 channels with
equal width in the dry state of particles. The meteorological data of $T$, RH and WS and of GRad
were measured by standardised sensors (HD52.3D17, Delta OHM, Italy, and SMP3
pyranometer, Kipp and Zonen, the Netherlands, respectively) with a time resolution of 1 min.



The DMPS and meteorological measurements were accomplished at the Budapest platform for
Aerosol Research and Training (BpART) Laboratory (N 47° 28' 29.9", E 19° 3' 44.6", 115 m
above mean sea level) of the Eötvös University (Fig. 1). The location represents a well-mixed,
average atmospheric environment for the city centre due to its geographical and meteorological
conditions (Salma et al., 2016a). The local emissions include diffuse urban traffic exhaust,
household/residential emissions and limited industrial sources together with some off-road
transport (Salma et al., 2020a). In some time intervals, long-range transport of air masses can
also play a role.

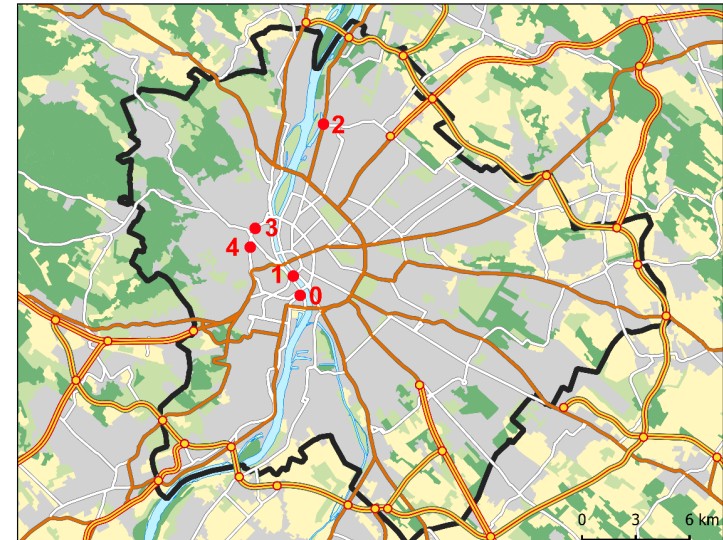

**Figure 1.** Location of the measurement sites in Budapest. 0: BpART Laboratory, 1: Szabadság Bridge,
2: Váci Road, 3: Széna Square and 4: Alkotás Road. The border of the city (in black colour), Danube
River are the major routes are also indicated.

The data of the criteria air pollutants were acquired from a measurement station of the National
Air Quality Network at Széna Square (Fig. 1) located in 4.5 km from the BpART Laboratory
in the upwind-prevailing direction (Salma and Németh, 2019). This station serves as a reference
for our long-term air quality-related research activities in several aspects, and proved to be
favourable for this purpose.

Atmospheric transport of chemical species was assessed through large-scale weather types. We
utilised macrocirculation patterns (MCPs), which were invented specifically for the Carpathian



Basin (Péczely, 1957; Maheras et al., 2018). The classification of the MCPs is based on the
position, extension and development of cyclones and anticyclones relative to the Carpathian
Basin considering the sea-level pressure maps constructed for 00:00 UTC in the North-
Atlantic–European region on a daily basis. A brief survey on the MCPs and the actual codes
for year 2020 utilised in the interpretations are given in Table S1 and Fig. S1, respectively in
the Supplement. The relative occurrences of the weather types in year 2020 were roughly in
line with multiple-year frequencies. Extended anticyclonic weather types usually indicate that
the air masses are stagnant, and that the importance of local or regional sources prevail above
the air transport from distant sources. Under cyclonic weather conditions and frontal systems,
the transported air masses can yield more pronounced effects.

Census of motor road vehicles was performed on three major routes and on a bridge over the
Danube River by the Budapest Public Roads Ltd. The measurement sites were on Szabadság
Bridge, Váci Road, Széna Square and Alkotás Road (Fig. 1), which are described in more detail
in the Supplement. The counting was based on permanent electronic devices with inductive
loops and passenger cars, high- and heavy-duty vehicles and buses were recorded in both
directions. The time resolution of the data was 1 h, and their coverage was >90% of all possible
item in a year. The sites cover a wide range of maximum hourly mean vehicle flow from about
1200 to 4600 h$^{-1}$. Szabadság Bridge has the smallest traffic intensity of the sites, but it proved
to be a very valuable microenvironment for the study since it is part of the internal boulevard.
The routes showed coherent and common aggregate time properties and, therefore, they are to
be proportional to general vehicular traffic flow in the city centre.
**2.2 Time intervals of interest**
Time intervals from 1 January to 31 July in 2017, 2018, 2019 and 2020 were studied. This
included all major measures related to the first outbreak of the Covid-19 pandemic in Budapest.
Within these seven months, five consecutive time intervals were selected for comparative
purposes: 1) from 1 January till the beginning of the state of emergency at 15:00 on 11 March,
which is referred as Pre-emergency phase, 2) from the beginning of the state of emergency to
27 March (till the beginning of the restriction on movement), which is called here Pre-
restriction phase, 3) from the beginning of the restriction on movement till its end in Budapest
on 17 May, which is denoted as Restriction phase, 4) from the end of the restriction on
movement in Budapest till the end of the state of emergency on 17 June, which is referred as





Post-restriction phase and 5) from the end of the state of emergency till 31 July, which is called
Post-emergency phase. An overview on the pandemic phases with further details of possible
relevance for air quality is summarised in Fig. S1. Equivalent time intervals in years 2017–
2019, which correspond to these phases were considered for comparative evaluation purposes.

Local daylight saving time (LDST=UTC+1 or UTC+2) was chosen as the time base for the
atmospheric concentrations and road traffic data because it had been observed that the daily
activity time patterns of inhabitants largely influences these variables in cities (Salma et al.,
2014). The meteorological data were expressed in UTC+1 since their temporal behaviour is
primarily controlled by natural processes.
**2.3 Data treatment and modelling**
Medical studies with the influenza virus indicated that absolute humidity (AH) constrains both
transmission efficiency and virus survival more than RH (Shaman and Kohn, 2009). For this
reason, the hourly mean RH values (%) were converted to AH (g m$^{-3}$) by a practical form of
Clausius-Clapeyron equation:

$$\mathrm{AH} = \frac{e(T_0) \times \exp\left(A \times \frac{T}{T+B}\right) \times \mathrm{RH} \times C}{T+273.15},$$    (1)

where $T$ is expressed in °C, $e(T_0)$=6.112 hPa is the saturation vapour pressure at $T_0$=0 °C,
$A$=17.67, $B$=243.5 °C and $C$=2.167 (WMO, 2008).

Vertical transfer of gases and aerosol particles emitted or generated at the Earth surface can
largely be affected by the dynamics of the PBLH. It causes dilution of pollutants by mixing.
The PBLH data were obtained from the 5th generation of the European Center of Medium
Range Weather Forecasting (ECMWF) atmospheric reanalysis (ERA5) database using
Copernicus Climate Change Service (C3S, 2017). ERA5 combines the CY41R2 version of the
ECMWF's Integrated Forecast System model data on 137 hybrid sigma vertical levels with
newly available observations assimilated at every hour. In the present study, the daily
maximum PBLH values (PBLH$_{max}$) were regarded to be proportional with the mixed air
volumes.


The data with a time resolution of smaller than 1 h were averaged for 1 h. The coverage of the
hourly data was typically above 90% of all possible items in each year. Descriptive statistics,
thus count, minimum, median, maximum, geometric mean with standard deviation (SD) of all
variables were derived for the time interval studied and its each pandemic phase in year 2020
(Y2020). The characteristics were compared to the corresponding data in an average reference
year (Y3Ref). This contains averages of the parallel hourly mean data of the three years 2017–
2019. Longer time span than this would not necessarily be advantageous since some chemical
species in Budapest show tendentious change on a scale of ten years (Mikkonen et al., 2020)
and the urban traffic could also be changed. Comparative evaluations are sometimes performed
via relative change (RDiff) of medians ($m$) derived for a selected time span, which can be
described as

$$\text{RDiff} = \frac{m(\text{Y2020}) - m(\text{Y3Ref})}{m(\text{Y3Ref})}. \tag{2}$$

In our case, the time spans considered were the intervals of the five pandemic phases in both
Y2020 and Y3Ref. The quantity RDiff essentially expresses the ratio of medians. It is very
important to stress immediately that the ratios are largely influenced by the absolute magnitude
of variables and could be misleading if interpreted alone. In addition, different variables can
have very different ranges of variability. A better metric could, therefore, be standardised
anomaly (SAly), which is defined as

$$\text{SAly} = \frac{m(\text{Y2020}) - m(\text{Y3Ref})}{\text{SD}}. \tag{3}$$

For GRad, which evolves daily from their very low values overnight in a large number, which
were not considered, the anomaly was not standardised to its (expanded) SD, but instead, it
was calculated simply as a difference $m(\text{Y2020}) - m(\text{Y3Ref})$ in its absolute unit.

A change in fluctuating and periodically varying concentrations (see Sect. 1) over a pandemic
phase in Y2020 was quantified to be significant with respect to the equivalent interval of Y3Ref
if both their RDiff and SAly metrics were significant. This is specified further in Sect. 3.5.

Average diurnal variations of all variables for workdays and holidays over each pandemic
phase in the average reference year 2017–2019 and year 2020 were calculated by selecting all



individual data for a particular hour of day on workdays or on holidays over the time interval
under evaluation and by averaging them.

Potential differences in the spatial distributions of the chemical species of interest during each
pandemic phase were studied through the surface concentrations downloaded from the
Copernicus Atmosphere Monitoring Service (CAMS) with a grid resolution of 0.1°×0.1°.
Reanalysed concentrations were based on seven state-of-the-art European models (CHIMERE,
EMEP, EURAD-IM, LOTOS-EUROS, MATCH, MOCAGE and SILAM). The system
provides daily 96-h forecasts with hourly outputs of several chemical species. The hourly
analysis at the Earth surface is done a posteriori for the past day using a selection of
representative air quality data from European monitoring stations (Marécal et al., 2015; CAMS,

2019).

**3 Results and discussion**
The changes in atmospheric concentrations are interpreted after the effects of the confound
variability in local meteorological conditions and in (long-range) transport of atmospheric air
masses are evaluated and quantified.
**3.1 Meteorological conditions**
The hourly average meteorological data over the time interval considered were in line with
ordinary characteristics measured at the BpART Laboratory (Salma and Németh, 2019;
Mikkonen et al., 2020). The $T$ in 2020 was colder by 0.4 °C than in the average reference year,
and the relative differences for median RH, WS, GRad and $PBLH_{max}$ were −3, −8, +3 and
+15%, respectively. These alterations, except for the $PBLH_{max}$, are not significant (remained
within ±10%). There were, however, two important alterations from the multiple-years'
weather situations. Spring 2020 was extraordinary dry; it was the third driest season since 1901.
The drought started after 7 March and continued in April and May, and finally, it was followed
by frequent, continued and spatially extended rains in June (Fig. S1). The number of foggy
hours (160) based on the measurements at the Budapest Liszt Ferenc International Airport in
January 2020 was more than four times larger than in the average reference year.

An overview on the major meteorological data during the whole state of emergency interval
(98 days) is summarised in Table S2. The drought did not seem to influence substantially the



WS and GRad but affected considerably the RH and indirectly the $PBLH_{max}$. The alterations in
the $PBLH_{max}$ in the average reference year and year 2020 over the pandemic phases are,
therefore, quantified separately in Table 1 and are shown in Fig. S2. Time series for WS and $T$
are also given in Figs. S3 and S4, respectively.

**Table 1.** Medians of the daily maximum planetary boundary layer height (km) in the average reference
year of 2017–2019 (Y3Ref) and year 2020 (Y2020) together with their relative difference (RDiff) in %
and their anomaly standardised to SD (SAly) over the five phases of the first outbreak of the COVID-
19 pandemic.

| Pandemic phase | Y3Ref | Y2020 | RDiff | SAly |
|---|---|---|---|---|
| Pre-emergency | 0.66 | 0.88 | +32 | +0.4 |
| Pre-restriction | 1.4 | 1.4 | +1 | +0.0 |
| Restriction | 1.5 | 1.8 | +18 | +0.5 |
| Post-restriction | 1.6 | 1.3 | –21 | –0.7 |
| Post-emergency | 1.8 | 1.7 | –8 | –0.3 |


It is seen in Fig. S2 that the Restriction phase – which is of particular interest – was influenced
by the $PBLH_{max}$ in a more-or-less persistent manner without larger oscillations or fluctuations.
The RDiff properties are taken into consideration when quantifying the concentration changes
(Sect. 3.5).
**3.2 Motor vehicle road traffic**
Time series of vehicle flow on a major route (Váci Road) over the time interval studied in the
average reference year and year 2020 are shown in Fig. 2 as examples. The other urban sites
exhibited very similar time behaviour and tendencies.

The time variations for vehicle flow showed a clear periodicity. On each workday, two peaks
– corresponding to the early morning and late afternoon rush hours – were detected. In addition
to this periodicity, the smoothed curves also revealed an obvious cycling due to repeated
workdays and holidays sequence. More importantly, the time series implied that in the Pre-
emergency pandemic phase, the road traffic in the city centre in Y2020 was very similar to that
in Y3Ref. The differences only appeared as shifts in time, which were caused by the occurrence
of holidays in the average reference year and year 2020. Two weeks before the introduction of



the restriction on movement, the vehicle circulation started already declining, and in the last
week of the Pre-restriction phase, it already reached the level observed later in the Restriction
phase. During these eight weeks, the vehicular circulation on workdays was around the
ordinary levels on holidays in 2017–2019. The circulation approached its ordinary values only
after the first week of the Post-restriction phase step wisely and reached it around 2 June 2020.
After that time, the curves for the two years were at almost identical levels again. The changes
in the vehicle flow are quantified in Sect. 3.5 together with the pollutant concentrations.

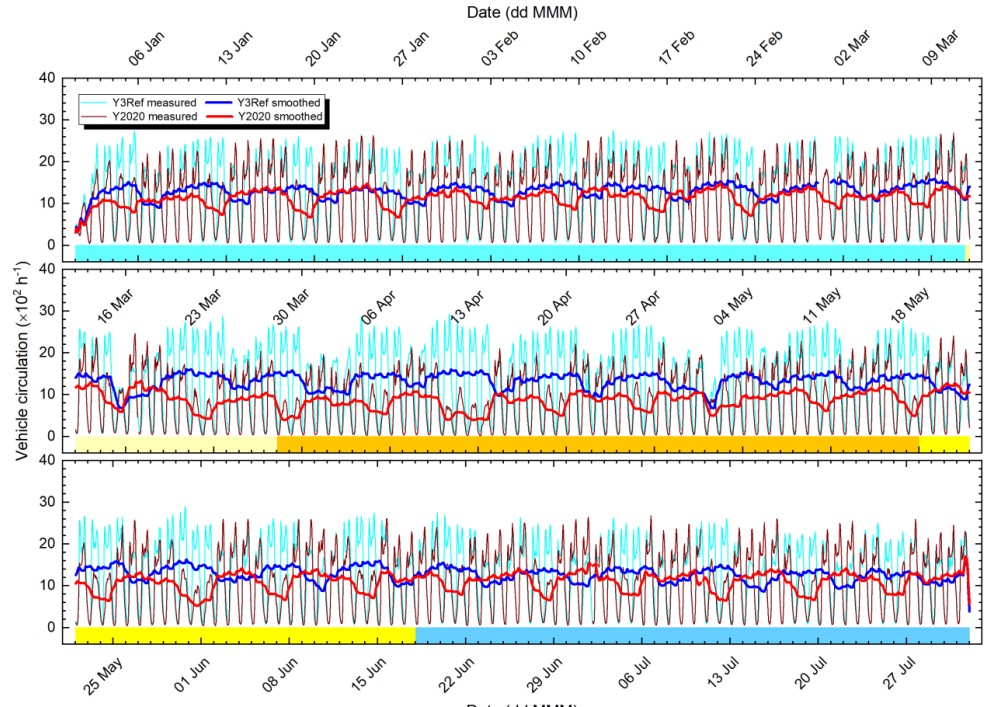

**Figure 2.** Time variation of motor vehicle circulation on a major route (Váci Road) in Budapest in both directions in the average reference year of 2017–2019 (Y3Ref) and year 2020 together with their 24-h smoothed curves over the five consecutive phases of the first COVID-19 outbreak. The phases are marked by the following colour codes: Pre-emergency phase lighter blue, Pre-restriction phase lighter yellow, Restriction phase orange, Post-restriction phase darker yellow and Post-emergency phase darker blue. The tick labels of the abscissa indicate the Mondays in 2020.

Average diurnal variations of vehicle traffic on Váci Road separately for workdays and
holidays over the different pandemic phases are given in Fig. S5 as examples. On workdays,
the traffic increased monotonically from 05:00 to 08:00 and reached its first maximum between





08:00 and 09:00. The circulation remained at an elevated level until the second maximum
which appeared between 17:00 and 18:00. The vehicle flow decreased monotonically after the
second peak until ca. 04:00 on next morning. The shapes of the curves on holidays were
different from that on workdays. The former curve exhibited slower elevation in the morning,
its first maximum was shifted toward noon and formed a wider plateau from approximately
11:00 to 19:00. The mean traffic from 00:00 to 05:00 was greater on holidays than on workdays.
This all was caused by different daily activities and habits of inhabitants on workdays and
holidays.

The shapes of the diurnal patterns remained virtually unchanged when the average reference
year and year 2020 were compared. A small difference could be identified in Pre-restriction
and Restriction pandemic phases between 16:00 and 19:00, when the traffic flow on weekends
seemed to be systematically lower in excess in year 2020 than in the average reference year.
This could be due to the limitations on shopping and to modified going out routines of
inhabitants under the restrictions. Similarly, the early morning peak on workdays in the Post-
emergency (and partly in the Post-restriction) phases was smaller in excess in Y2020 than that
in Y3Ref, which can likely be linked to less people going physically to work due to propagated
home-office jobs.
**3.3 Time series of concentrations**
Time series of NO, $O_3$, $PM_{2.5}$ mass and $N_{6-1000}$ atmospheric concentrations over the time
interval studied are shown in Figs. 3–6, respectively. The chemical species selected represent
primary pollutant gases, secondary pollutant gases and two different aerosol properties,
respectively. The corresponding curves for $NO_2$, CO, $SO_2$, $PM_{10}$ mass and $N_{100-1000}$ are
displayed in Figs. S6–S10, respectively. The overall character of the smoothed 7-month curves
are in line with the distributions of the monthly median concentrations of the species at an
identical location over several years (Salma et al., 2020b). The time series showed both certain
similarities and differences when they were compared.

The curves for both measured and smoothed data demonstrated that the concentrations varied
substantially in time. The changes on the smoothed curves seemed to be fluctuations, while the
data series possessed diurnal periodicity as well. The overall relative SDs (RSDs) for NO, $NO_2$,



CO, O$_3$, SO$_2$, PM$_{10}$ mass, PM$_{2.5}$ mass, $N_{6-1000}$, $N_{6-100}$, $N_{25-100}$ and $N_{100-1000}$ in years 2017–2019
were 115, 56, 43, 91, 37, 56, 74, 63, 69, 68 and 68%, respectively (cf. Sect. 1).

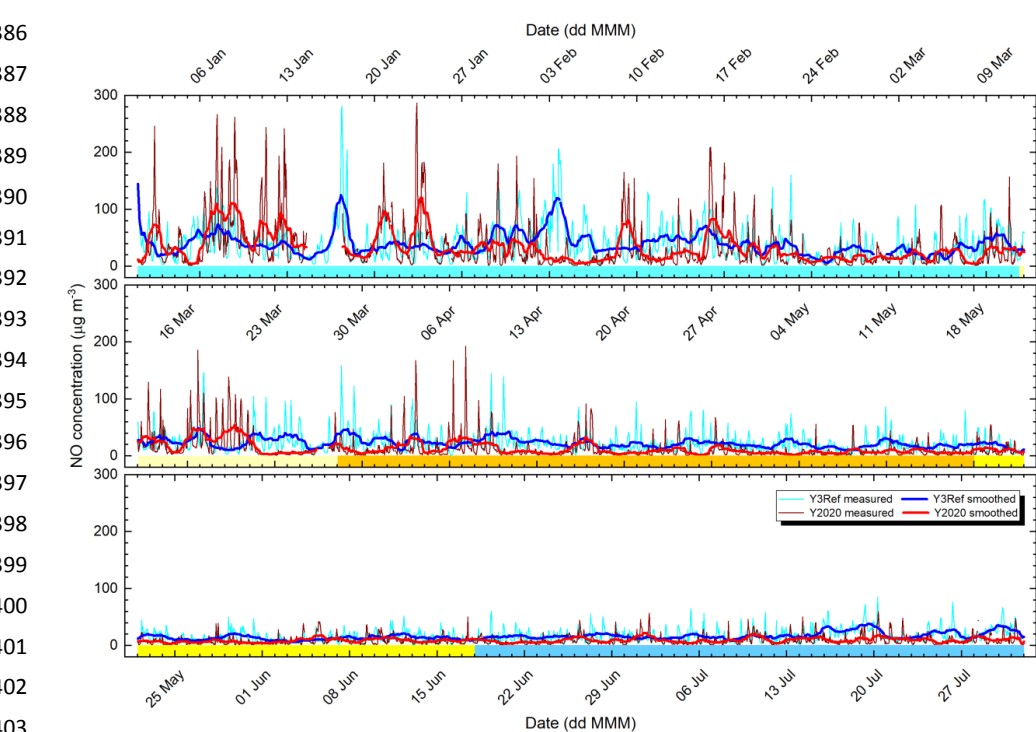

**Figure 3.** Time variation of NO concentration in the average reference year of 2017–2019 (Y3Ref) and
year 2020 together with their 24-h smoothed curves over the five consecutive phases of the first
COVID-19 outbreak. The phases are marked by the following colour codes: Pre-emergency phase
lighter blue, Pre-restriction phase lighter yellow, Restriction phase orange, Post-restriction phase darker
yellow and Post-emergency phase darker blue. The tick labels of the abscissa indicate the Mondays in

2020.

The time distributions of monthly mean RSDs were complex. For species, which do not show
seasonal trend such as particle number concentrations and perhaps PM$_{10}$ mass, the distributions
of monthly RSDs were also featureless. For SO$_2$, which tends to exhibit smaller concentration
levels in summer than in winter, the distribution of its monthly RSDs seemed to have an
opposite behaviour. For O$_3$, which exhibits larger concentration levels in summer than in
winter, the distribution of monthly RSDs showed again an opposite behaviour. These
relationships are in accordance with general metrological expectations. Excitingly, for NO,
NO$_2$, CO and perhaps PM$_{2.5}$ mass, the distributions of monthly RSDs appeared to follow
roughly the concentration trends in parallel within the concentration ranges actually measured.
The largest decrease in the RSDs from winter to summer was observed for NO, which was
approximately 20% (of its annual mean RSD). The latter association could likely be linked to
meteorological conditions and source/sink intensities of these pollutants.

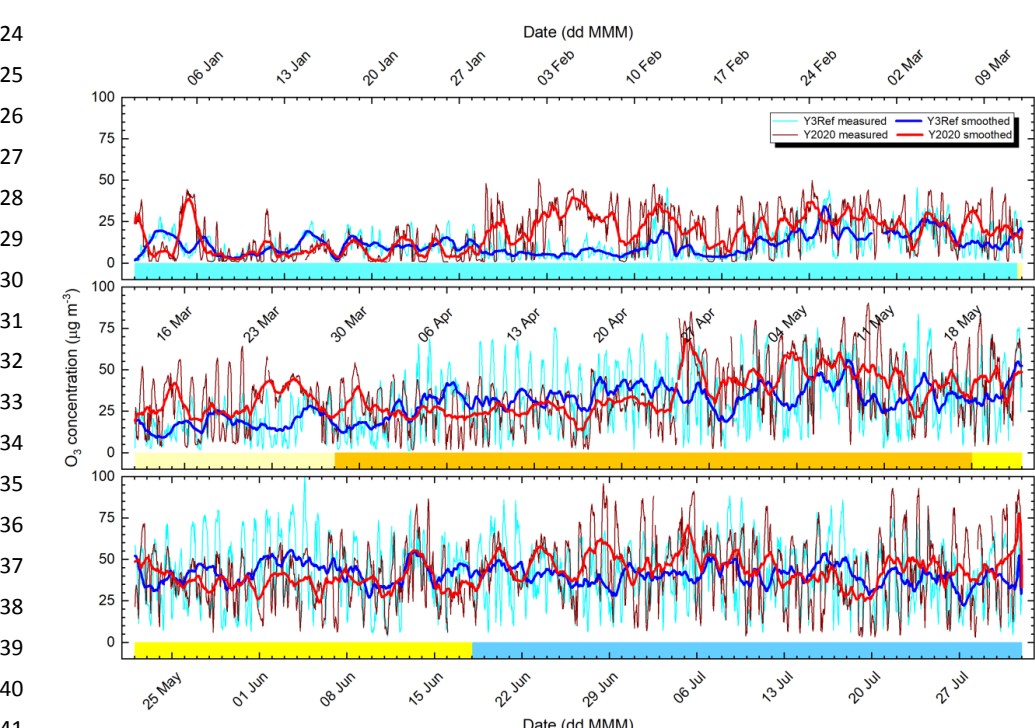

**Figure 4.** Time variation of $O_3$ concentration in the average reference year of 2017–2019 (Y3Ref) and year 2020 together with their 24-h smoothed curves over the five consecutive phases of the first COVID-19 outbreak. The phases are marked by the following colour codes: Pre-emergency phase lighter blue, Pre-restriction phase lighter yellow, Restriction phase orange, Post-restriction phase darker yellow and Post-emergency phase darker blue. The tick labels of the abscissa indicate the Mondays in 2020.

Many chemical species investigated originate from rather different sources. Nevertheless, their
atmospheric concentrations often changed coherently, particularly in winter and early spring.
A nice example is the interval of approximately 14–28 March 2020 when most species varied
consistently. The MCPs for these days indicate strong anticyclonic weather types over the
Carpathian Basin, stagnant and relatively calm meteorological conditions without precipitation
in the area (Fig. S1). It is a demonstration of the common effects of regional meteorology on
atmospheric concentrations. The strongest connections are related to cold air masses above the
basin which generate a lasting $T$ inversion and relatively shallow planetary boundary layer
(cold air pool). It confirms that the daily evolution of regional meteorology can have higher
influence on atmospheric concentrations than the source intensities under such conditions
(Salma et al., 2020a).


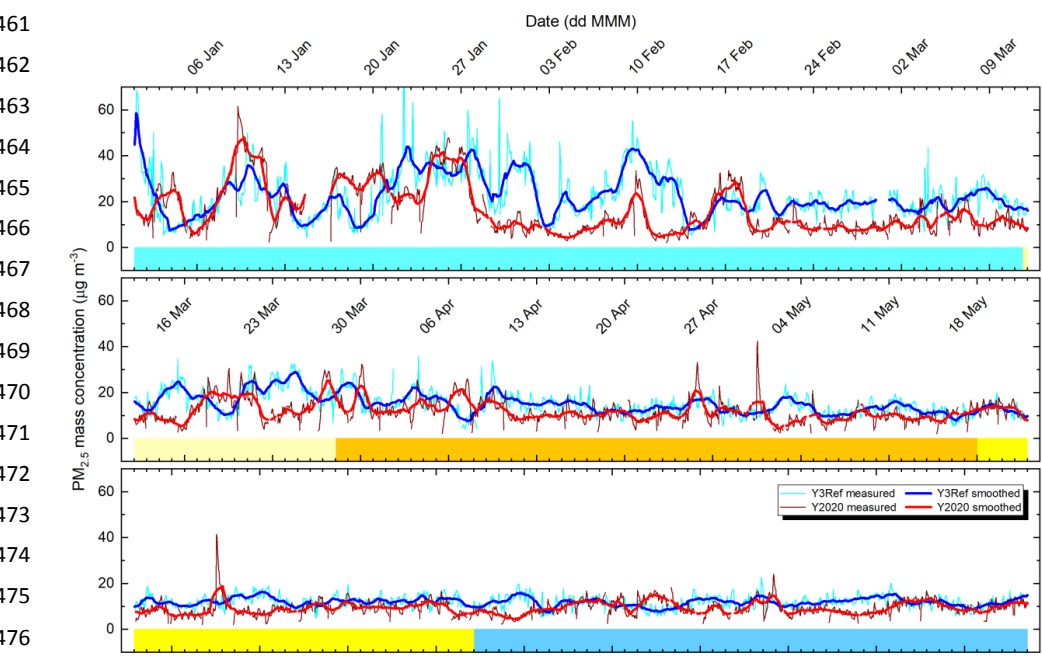

**Figure 5.** Time variation of PM$_{2.5}$ mass concentration in the average reference year of 2017–2019
(Y3Ref) and year 2020 together with their 24-h smoothed curves over the five consecutive phases of
the first COVID-19 outbreak. The phases are marked by the following colour codes: Pre-emergency
phase lighter blue, Pre-restriction phase lighter yellow, Restriction phase orange, Post-restriction phase
darker yellow and Post-emergency phase darker blue. The tick labels of the abscissa indicate the
Mondays in 2020.

The curves for PM$_{2.5}$ mass and $N_{6-1000}$ confirmed that there is week association between these
two types of aerosol metrics (de Jesus et al., 2019). They are related mainly via meteorological
properties, which is anyway active for all pollutants. It was sensible, therefore, that both types
of aerosol concentrations were included into the study as separate variables.



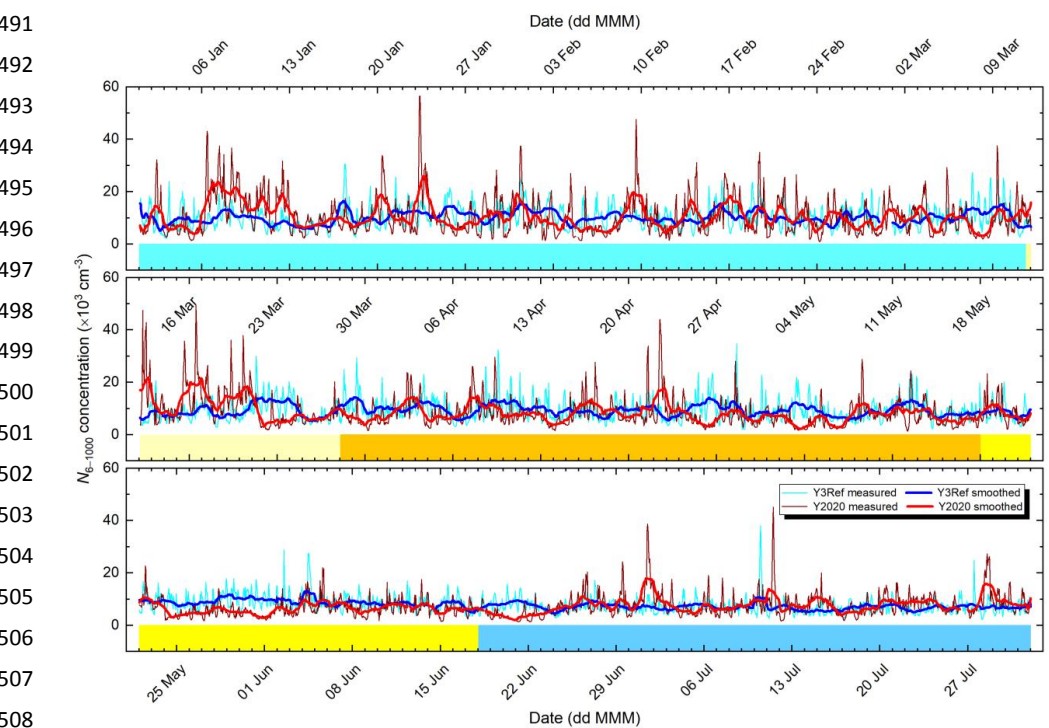

**Figure 6.** Time variation of $N_{6-1000}$ concentration in the average reference year of 2017–2019 (Y3Ref) and year 2020 together with their 24-h smoothed curves over the five consecutive phases of the first COVID-19 outbreak. The phases are marked by the following colour codes: Pre-emergency phase lighter blue, Pre-restriction phase lighter yellow, Restriction phase orange, Post-restriction phase darker yellow and Post-emergency phase darker blue. The tick labels of the abscissa indicate the Mondays in 2020.

### 3.4 Diurnal variability of concentrations

Average diurnal variations of NO, $O_3$, $SO_2$, $PM_{2.5}$ mass and $N_{6-100}$ separately for workdays and holidays over the Restriction pandemic phase, for which the differences in the shapes are expected to be the largest, are shown in Fig. 7 as examples.

The curves of NO and $N_{6-100}$ (together with $NO_2$, CO and $N_{6-1000}$, which were not shown) followed the typical pattern of road traffic (Fig. S5). They can largely be related to vehicular sources (tailpipe emissions, primary and secondary particles), and can advantageously be applied for assigning potential concentration changes to traffic reduction. The curves of $N_{6-100}$ contained in addition the characteristic midday peak, which is caused by atmospheric NPF



events. It is worth realising that its position was shifted to later time. There were only nine
quantifiable NPF events during the Restriction phase in year 2020, which might not result in a
representative shape. This should definitely be investigated and clarified in detail when the
necessary data sets and their treatment become available.

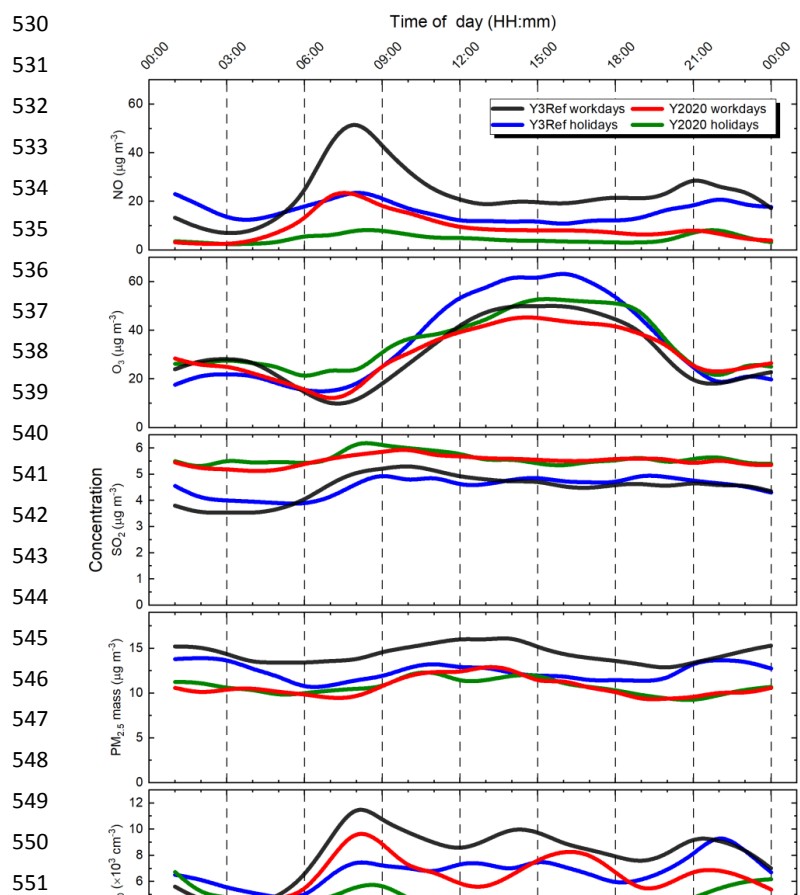

**Figure 7.** Average diurnal variations of NO, $O_3$, $SO_2$, $PM_{2.5}$ mass and $N_{6-100}$ concentrations separately
for workdays and holidays in the average reference year of 2017–2019 (Y3Ref) and year 2020 during
the Restriction phase of the first COVID-19 outbreak.





The curves for $O_3$ seemed to be opposite to NO and $NO_2$, which is in line with the
understanding of their reaction mechanism in volatile-organic-compound (VOC)-limited
chemical regime (Lelieveld and Dentener, 2000). It involves, for instance, aromatics such as
benzene and toluene that largely originate from traffic source. The shapes in Y2020 seemed to
be somewhat flattened, and they were also affected by the clock change.

The curves for $SO_2$ (together with $PM_{10}$ mass and $N_{100-1000}$, which were not shown) tracked the
traffic pattern very loosely. They could at most partially be related to traffic through diesel fuel,
dispersion and suspension of crustal rock, road dust and road surfaces by moving vehicles and
growth of emitted particles to larger sizes. There was no obvious connection between the traffic
and $PM_{2.5}$ mass, which confirms that fine particles in Budapest mainly originate from non-
vehicular sources.

**3.5 Quantification of concentration changes**


There are several mathematical statistical tests to determine whether atmospheric
concentrations over some time intervals in different years belong to the same distribution or
not. These methods, however, quantify the joint influence of all environmental contributions
(Sect. 1) as one and do not provide information on the causal relationships. The method
described and applied below allows to unfold some potential confounding effects of
environmental variables (e.g. PBLH) from concentration changes in order to gain a closer
insight into source intensities of motor vehicles.

Median concentrations of pollutant gases and aerosol particles, median traffic circulation data
together with their relative differences and standardised anomaly values for the five pandemic
phases in the average reference year and year 2020 are summarised in Tables 2–6. It should be
noted that the standardised anomalies are rather small when recalling, for instance, the rigorous
concept of the limits of detection (3×SD) and determination (10×SD) in analytical chemistry.
This is largely caused by the dynamic features of related atmospheric properties and processes
(Sects. 1. and 3.3).

We showed in Sect. 3.1 that the effect of $PBLH_{max}$ of the local meteorological conditions on
the atmospheric concentrations could be considerable, and, therefore, its influence was taken
into account. A change in median concentrations for a pandemic phase was quantified to be



significant if both its relative difference fell outside the band of $[\pm10–f_{mix}\times RDiff(PBLH_{max})]\%$
and its SAly was outside the range of $\pm0.3$. The multiplication factor $f_{mix}$ accounts for non-
homogeneous mixing of air constituents within boundary layer and for the effects of the daily
PBLH evolution. It was roughly estimated to be approximately 0.5. Its negative sign expresses
that atmospheric concentrations vary in a reciprocal manner with PBLH. The selected criteria
were based upon exercises with the data in the individual years 2017, 2018 and 2019. The
procedure represents a sensible and consequent approach, though alternative limits could also
be set.

The Pre-emergency phase (Table 2) fitted completely into the heating season. The traffic flows
in city centre were identical, except for Váci Road, where they were somewhat lower in Y2020
with respect to Y3Ref. They were mainly cause by some local traffic arrangements. The
$PBLH_{max}$ increased by 32%, which is substantial and could have influenced the concentrations.
NO, $NO_2$, CO, $PM_{10}$ mass, $PM_{2.5}$ mass and $N_{100–1000}$ decreased, while $O_3$ increased
substantially. Most of these changes were, however, insignificant. The exceptions were NO,
$O_3$, $PM_{10}$ mass and $PM_{2.5}$ mass, and the latter two exhibited the largest anomalies. These
species can have multiple sources. Organic matter and elemental carbon, for instance, make up
approximately 35% of the $PM_{2.5}$ mass in inter (Salma et al., 2020a). Biomass burning is the
major source of carbonaceous aerosol in this season with its relative contribution to total carbon
of approximately 67%. The share of fossil-fuel combustion sources is around 25%. This all
implies that $PM_{2.5}$ mass concentrations can fluctuate irregularly and largely due to their
sources.

The higher concentration of $O_3$ could partly be related to the lower concentrations of NO.
Ozone exhibits a strong seasonal dependency (Salma et al., 2020b). Generally lower
concentrations in winter and early spring can easily be disturbed or influenced by its non-linear
chemistry and by high WS values. Furthermore, the main differences in these concentrations
appear sporadically and in an isolated manner in their time series, and there was no coherence
among traffic-related variables. Therefore, all these significant variations were very likely
caused by other reasons than vehicle flow and resulted due to inter-annual variability in
emissions, formation processes, sinks and local meteorology.



**Table 2.** Median atmospheric concentrations of NO, $NO_2$ (both in units of µg m$^{-3}$) CO (mg m$^{-3}$), $O_3$,
$SO_2$, $PM_{10}$ mass, $PM_{2.5}$ mass (all in µg m$^{-3}$), $N_{6-1000}$, $N_{6-100}$, $N_{25-100}$, $N_{100-1000}$ (all in $10^3$ cm$^{-3}$) and median
vehicle road traffic (h$^{-1}$) on Szabadság Bridge, Váci Road, Széna Square and Alkotás Road in the
average reference year of 2017–2019 (Y3Ref) and year 2020 together with their relative difference
(RDiff, %) and their anomaly standardised to SD (SAly) for Pre-emergency phase of the first COVID-
19 outbreak. Chemical species with significant change are shown in bold.

| Int. | Variable | Y3Ref | Y2020 | RDiff | SAly |
|------|----------|-------|-------|-------|------|
| 1) Pre-emergency phase (71 days) | **NO** | 33 | 18 | −45 | −0.6 |
| | $NO_2$ | 67 | 50 | −26 | −0.7 |
| | CO | 0.74 | 0.58 | −21 | −0.8 |
| | **$O_3$** | 9.4 | 16 | +68 | +0.3 |
| | $SO_2$ | 5.5 | 5.4 | −1 | +0.0 |
| | **$PM_{10}$** | 45 | 29 | −36 | −1.2 |
| | **$PM_{2.5}$** | 21 | 12 | −42 | −1.2 |
| | $N_{6-1000}$ | 9.5 | 8.8 | −7 | −0.2 |
| | $N_{6-100}$ | 7.2 | 6.8 | −6 | −0.1 |
| | $N_{25-100}$ | 3.5 | 3.1 | −10 | −0.2 |
| | $N_{100-1000}$ | 2.2 | 1.7 | −21 | −0.5 |
| | Szabadság B. | 676 | 640 | −5 | −0.1 |
| | Váci R. | 1589 | 1299 | −18 | −0.4 |
| | Széna S. | 1374 | 1437 | +5 | +0.1 |
| | Alkotás R. | 2517 | 2425 | −4 | −0.1 |


The Pre-restriction phase (Table 3) was rather short (16 days), and, therefore, its interpretation
should be approached with special caution due to some issues in representativity. It was also
completely part of the heating season, and the extreme drought in the Carpathian Basin in 2020
could also play a role. The $PBLH_{max}$ as almost identical in both years (Fig. S2). In this pandemic
phase, the concentrations of $NO_2$, $PM_{2.5}$ mass and perhaps NO declined, while $O_3$ was
enhanced. Excitingly, CO did not show substantial decrease. They could likely be affected by
lower traffic circulation during its last half/week (Fig. 2).





**Table 3.** Median atmospheric concentrations of NO, $NO_2$ (both in units of µg m$^{-3}$) CO (mg m$^{-3}$), $O_3$,
$SO_2$, $PM_{10}$ mass, $PM_{2.5}$ mass (all in µg m$^{-3}$), $N_{6-1000}$, $N_{6-100}$, $N_{25-100}$, $N_{100-1000}$ (all in $10^3$ cm$^{-3}$) and median
vehicle road traffic (h$^{-1}$) on Szabadság Bridge, Váci Road, Széna Square and Alkotás Road in the
average reference year of 2017–2019 (Y3Ref) and year 2020 together with their relative difference
(RDiff) in % and their anomaly standardised to SD (SAly) for Pre-restriction phase of the first COVID-
19 outbreak. Chemical species with significant change are shown in bold.

| Int. | Variable | Y3Ref | Y2020 | RDiff | SAly |
|---|---|---|---|---|---|
| **2) Pre-restriction phase (16 days)** | NO | 20 | 12 | −39 | −0.3 |
| | **NO$_2$** | 57 | 45 | −22 | −0.5 |
| | CO | 0.60 | 0.56 | −8 | −0.2 |
| | **O$_3$** | 18 | 33 | +80 | +0.7 |
| | SO$_2$ | 5.1 | 5.4 | +7 | +0.3 |
| | PM$_{10}$ | 34 | 30 | −12 | −0.3 |
| | **PM$_{2.5}$** | 19 | 13 | −32 | −0.8 |
| | $N_{6-1000}$ | 8.1 | 8.4 | +4 | +0.1 |
| | $N_{6-100}$ | 6.7 | 6.9 | +4 | +0.1 |
| | $N_{25-100}$ | 2.9 | 3.2 | +9 | +0.1 |
| | $N_{100-1000}$ | 1.4 | 1.6 | +11 | +0.2 |
| | Szabadság B. | 652 | 417 | −36 | −0.8 |
| | Váci R. | 1522 | 939 | −38 | −0.7 |
| | Széna S. | 1371 | 1001 | −27 | −0.5 |
| | Alkotás R. | 2792 | 1925 | −31 | −0.7 |


The beginning one-third part of the Restriction phase (Table 4) fell into the heating season, and
the phase was fully incorporated into the extremely dry weather season. The vehicle flows were
reduced by approximately half uniformly at all locations. Concentrations of NO, $NO_2$, CO,
$PM_{2.5}$ mass, $N_{6-1000}$ and $N_{6-100}$ also changed significantly, and they all declined. The alterations
happened in a systematic or continuous manner in time (Figs. 2, 3, 5, 6, S6 and S7). These
species can be associated with vehicular road traffic. Except for $PM_{2.5}$ mass, which can be
mostly linked to household sources. At the same time, some other important pollutants such as
$N_{100-1000}$ or $SO_2$ – which are typically related to larger spatial extent or region and which could,
therefore, be influenced strongly by meteorology – did not change significantly. Similar
reductions were reported for other urban locations in the world (Keller et al., 2020; Le et al.,
2020; Lee et al., 2020; Tobías et al., 2020). This all can be interpreted that the alterations in
NO, $NO_2$, CO, $N_{6-1000}$ and $N_{6-100}$ concentrations were primarily caused by the lower vehicular





traffic intensity in the city, and that the effect of the PBLH could also contribute by
approximately 9% in an absolute sense.

**Table 4.** Median atmospheric concentrations of NO, NO$_2$ (both in units of µg m$^{-3}$) CO (mg m$^{-3}$), O$_3$,
SO$_2$, PM$_{10}$ mass, PM$_{2.5}$ mass (all in µg m$^{-3}$), $N_{6-1000}$, $N_{6-100}$, $N_{25-100}$, $N_{100-1000}$ (all in 10$^3$ cm$^{-3}$) and median
vehicle road traffic (h$^{-1}$) on Szabadság Bridge, Váci Road, Széna Square and Alkotás Road in the
average reference year of 2017–2019 (Y3Ref) and year 2020 together with their relative difference
(RDiff) in % and their anomaly standardised to SD (SAly) for Restriction phase of the first COVID-19
outbreak. Chemical species with significant change are shown in bold.

| Int. | Variable | Y3Ref | Y2020 | RDiff | SAly |
|---|---|---|---|---|---|
| 3) Restriction phase (51 days) | **NO** | 19 | 6.0 | −68 | −0.5 |
| | **NO$_2$** | 55 | 30 | −46 | −1.0 |
| | **CO** | 0.58 | 0.43 | −27 | −0.8 |
| | O$_3$ | 31 | 35 | +13 | +0.2 |
| | SO$_2$ | 5.4 | 5.5 | +3 | +0.1 |
| | PM$_{10}$ | 32 | 28 | −13 | −0.3 |
| | **PM$_{2.5}$** | 14 | 11 | −22 | −0.4 |
| | **$N_{6-1000}$** | 8.8 | 6.7 | −24 | −0.5 |
| | **$N_{6-100}$** | 7.4 | 5.3 | −28 | −0.6 |
| | $N_{25-100}$ | 3.2 | 2.8 | −12 | −0.2 |
| | $N_{100-1000}$ | 1.3 | 1.2 | −5 | +0.1 |
| | Szabadság B. | 689 | 318 | −54 | −1.2 |
| | Váci R. | 1626 | 803 | −51 | −1.0 |
| | Széna S. | 1537 | 844 | −45 | −1.0 |
| | Alkotás R. | 3031 | 1516 | −50 | −1.1 |


In the Post-restriction phase (Table 5) the vehicle flow recovered step wisely. The PBLH$_{max}$ in
Y2020 decreased substantially relative to Y3Ref (Table 1). Most chemical species such as NO$_2$,
SO$_2$, PM$_{10}$ mass, PM$_{2.5}$ mass, $N_{6-1000}$, $N_{6-100}$, $N_{25-100}$ and $N_{100-1000}$ exhibited significant changes.
The list included variables which often characterize the regional scale. At the same time, some
typical vehicular-related species such as NO and CO – which are not really water soluble –
were not among them. Most of the significant changes showed decreasing tendency, except for
SO$_2$ which increased. This was caused by a continuously increasing SO$_2$ concentration level
(Fig. S8), recorded at the other air quality monitoring stations in Budapest as well, and which
suggests that the increase was likely caused by temporal local source in the upwind direction
from the city as a disturbance. This all suggests that the alterations were mainly produced by


arrival of the continued and spatially extended rains in its second half (Fig. S1), which washed
out many chemical species from the urban and regional atmospheres. The quantification of the
effects of vehicular traffic on the air quality was not feasible under such conditions. This time
interval unambiguously demonstrates that the regional weather can cause similar changes in
the atmospheric concentrations as a very substantially (by 50%) reduced vehicle traffic.

**Table 5.** Median atmospheric concentrations of NO, $NO_2$ (both in units of µg m$^{-3}$) CO (mg m$^{-3}$), $O_3$,
$SO_2$, $PM_{10}$ mass, $PM_{2.5}$ mass (all in µg m$^{-3}$), $N_{6-1000}$, $N_{6-100}$, $N_{25-100}$, $N_{100-1000}$ (all in $10^3$ cm$^{-3}$) and median
vehicle road traffic (h$^{-1}$) on Szabadság Bridge, Váci Road, Széna Square and Alkotás Road in the
average reference year of 2017–2019 (Y3Ref) and year 2020 together with their relative difference
(RDiff) in % and their anomaly standardised to SD (SAly) for Post-restriction phase of the first COVID-
19 outbreak. Chemical species with significant change are shown in bold.

| Int. | Variable | Y3Ref | Y2020 | RDiff | SAly |
|---|---|---|---|---|---|
| 4) Post-restriction phase (31 days) | NO | 12 | 6.4 | –44 | –0.2 |
| | **$NO_2$** | 47 | 29 | –38 | –0.7 |
| | CO | 0.48 | 0.42 | –13 | –0.3 |
| | $O_3$ | 42 | 37 | +11 | –0.2 |
| | **$SO_2$** | 4.7 | 5.9 | +26 | +1.0 |
| | **$PM_{10}$** | 29 | 21 | –28 | –0.6 |
| | **$PM_{2.5}$** | 12 | 9.3 | –24 | –0.4 |
| | **$N_{6-1000}$** | 8.2 | 6.0 | –27 | –0.5 |
| | **$N_{6-100}$** | 6.8 | 4.9 | –27 | –0.5 |
| | **$N_{25-100}$** | 3.3 | 2.4 | –27 | –0.5 |
| | **$N_{100-1000}$** | 1.3 | 1.0 | –22 | –0.3 |
| | Szabadság B. | 670 | 575 | –14 | –0.3 |
| | Váci R. | 1536 | 1137 | –26 | –0.5 |
| | Széna S. | 1540 | 1387 | –10 | –0.2 |
| | Alkotás R. | 2597 | 2281 | –12 | –0.2 |


In the Post-emergency phase (Table 6), the traffic was at its ordinary level and there were no
larger weather alternations. Most concentrations – including some major vehicle-related
constituents such as CO and $N_{6-100}$ – did not change significantly. The exceptions were NO,
$NO_2$, $O_3$ and $PM_{2.5}$ mass. The first three variables are related to each other through atmospheric
chemistry. The changes can likely be explained by ordinary (inter-annual) variability in
sources, sinks, atmospheric transformation and transport.





**Table 6.** Median atmospheric concentrations of NO, $NO_2$ (both in units of µg m$^{-3}$) CO (mg m$^{-3}$), $O_3$,
$SO_2$, $PM_{10}$ mass, $PM_{2.5}$ mass (all in µg m$^{-3}$), $N_{6-1000}$, $N_{6-100}$, $N_{25-100}$, $N_{100-1000}$ (all in 10$^3$ cm$^{-3}$) and median
vehicle road traffic (h$^{-1}$) on Szabadság Bridge, Váci Road, Széna Square and Alkotás Road in the
average reference year of 2017–2019 (Y3Ref) and year 2020 together with their relative difference
(RDiff) in % and their anomaly standardised to SD (SAly) for Post-emergency phase of the first
COVID-19 outbreak. Chemical species with significant change are shown in bold.

| Int. | Variable | Y3Ref | Y2020 | RDiff | SAly |
|---|---|---|---|---|---|
| 5) Post-emergency phase (44 days) | **NO** | 16 | 7.4 | −54 | −0.3 |
| | **NO$_2$** | 51 | 32 | −37 | −0.7 |
| | CO | 0.43 | 0.42 | −2 | −0.1 |
| | **O$_3$** | 39 | 46 | +17 | +0.3 |
| | SO$_2$ | 4.0 | 4.3 | +9 | +0.3 |
| | PM$_{10}$ | 26 | 22 | −15 | −0.3 |
| | **PM$_{2.5}$** | 12 | 9.1 | −22 | −0.3 |
| | $N_{6-1000}$ | 6.7 | 6.7 | +0 | +0.0 |
| | $N_{6-100}$ | 5.5 | 5.4 | −2 | +0.0 |
| | $N_{25-100}$ | 2.7 | 2.8 | +5 | +0.1 |
| | $N_{100-1000}$ | 1.1 | 1.2 | +9 | +0.1 |
| | Szabadság B. | 690 | 663 | −4 | −0.1 |
| | Váci R. | 1471 | 1218 | −17 | −0.3 |
| | Széna S. | 1594 | 1511 | −5 | −0.1 |
| | Alkotás R. | 2507 | 2531 | +1 | +0.0 |

**3.6 Change rates**
Linear regression analysis between the median RDiff for vehicle traffic in the city centre on
one side and RDiff for selected chemical species on the other side in the pandemic phases
yielded change rates and SDs for NO, $NO_2$, $N_{6-1000}$ and CO were 0.63±0.23, 0.57±0.14,
0.40±0.17 and 0.22±0.08, respectively. For $PM_{10}$ mass and $PM_{2.5}$ mass, the rates were slightly
negative and insignificant. The latter two species are not really related to vehicle traffic in
Budapest. The data for the Post-restriction phase – which were very substantially affected by
local meteorology such as precipitation and frontal weather systems (Sect. 3.5 and Fig. S1) –
were excluded from this analysis. The change rates suggest that relative changes of nitrogen-
oxides with traffic is the most sensitive, total particle number concentration shows considerable
dependency, while variation of CO with traffic is modest. This is linked to their residence times
as well. The PM mass concentrations do not appear to be closely related with traffic intensity.





### 3.7 Spatial gradients

Spatial distributions of NO and $O_3$ in 2018–2019 and 2020 during the Restriction pandemic phase are shown in Figs. 8 and 9 as examples. The absolute concentrations can be different from the measured values due to the specialities and differences in the applied model, while the relative tendencies are expected to be expressed correctly. Figure 8 indicates that the differences from the corresponding median (spatial gradients) in 2020 were larger than in the reference year. This can be explained if the relative concentration changes at the outer parts of the city or near-city background were larger than in the centre. The spatial distribution of $NO_2$ was similar to NO, although its gradients were smaller than for NO. Spatial distributions of CO and $PM_{2.5}$ mass were featureless and similar to each other in 2018–2019 and 2020.

Spatial distributions of $O_3$ (Fig. 9) and, perhaps $SO_2$, exhibited relative decrease in the centre, which gradients were relatively small and similar to each other for both years. This all is in line with the tendencies observed in their measured concentrations (Sects. 3.3 and 3.4). We are aware that several pollutants originate from local or diffusive line sources, which can be enriched along roads and, therefore, much larger concentration gradients can occur on smaller spatial scales.

### 4 Conclusions

The relationships between urban air quality and motor vehicle road traffic are not straightforward since the contributions of traffic flow to pollutants concentrations are superimposed in the variability in local meteorological conditions, long-range transport of air masses and other sources/sinks. We introduced here an approach based on both relative differences and standardised anomalies, which helps unfolding some important confounding environmental factors. It can support creating a generalised picture on urban atmospheres.

The method was deployed on the Budapest data during the different phases of the first COVID-19 outbreak. Various restriction measures introduced due to the pandemic resulted in a decline of vehicle road traffic down to approximately 50% during the severest limitations. In parallel, concentrations of NO, $NO_2$, CO, $N_{6–1000}$ and $N_{6–100}$ decreased substantially, some other species

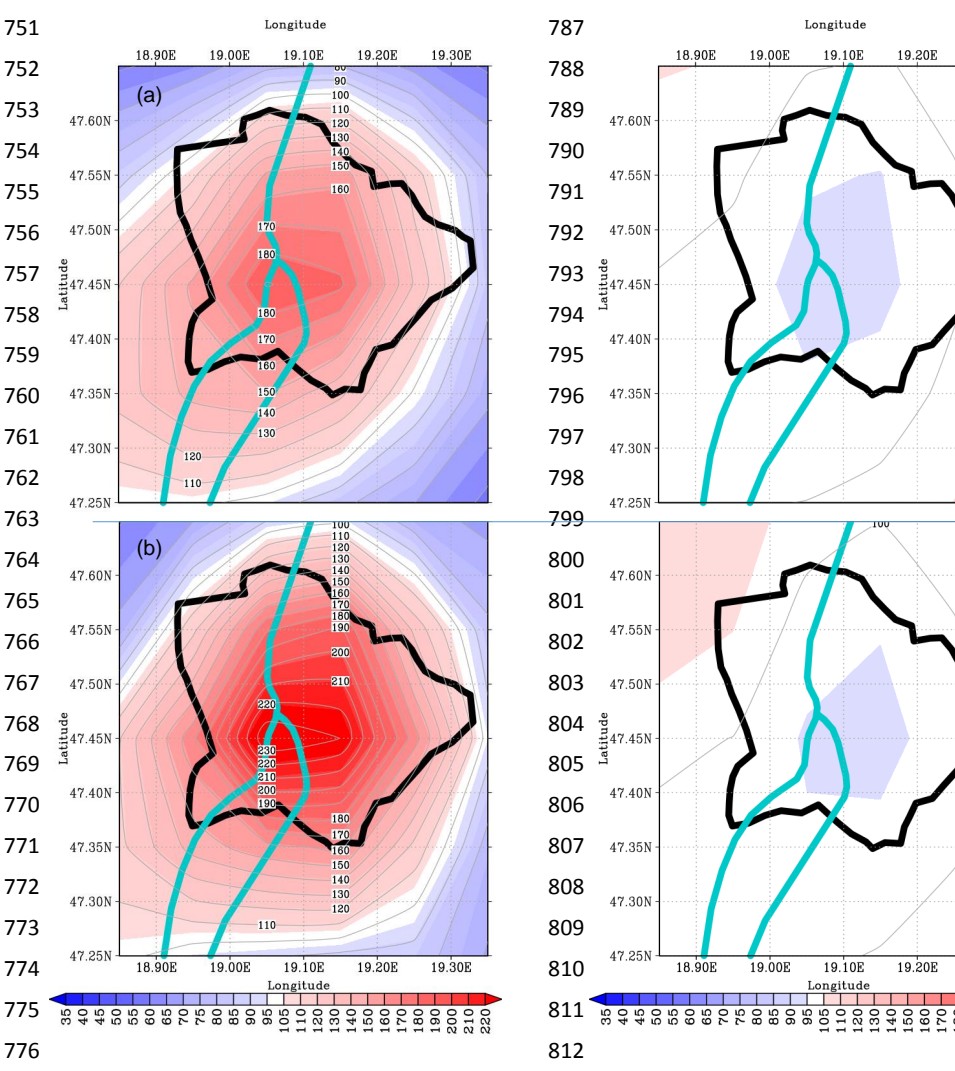

**Figure 8.** Spatial distribution of median NO concentration in Budapest in 2018–2019 (a) and 2020 (b) during the Restriction phase of the first COVID-19 outbreak. The concentrations were normalised to the overall spatial median concentrations of 0.93 and 0.59 µg m$^{-3}$, respectively. The border of the city and the Danube River are indicated with curves in black and blue colour, respectively for better orientation.

**Figure 9.** Spatial distribution of median O$_3$ concentration in Budapest in 2018–2019 (a) and 2020 (b) during the Restriction phase of the first COVID-19 outbreak. The concentrations were normalised to the overall spatial median concentrations of 60 and 67 µg m$^{-3}$, respectively. The border of the city and the Danube River are indicated with curves in black and blue colour, respectively for better orientation.



such as SO$_2$, PM$_{2.5}$ mass, PM$_{10}$ mass and $N_{100–1000}$ changed modestly or inconclusively, while
O$_3$ showed an increasing tendency. Change rates of NO and NO$_2$ with relative change of traffic
intensity (formally expressed as %/%) were the largest (approximately 0.6), total particle
number concentration showed considerable dependency (0.4), while variation of CO was
modest (0.2). Particulate matter mass concentrations, which are the most critical pollutant in
many European cities including Budapest, did not appear to be related with urban traffic. It was
demonstrated that a similar decrease in concentrations as observed in the strictest pandemic
phase can also be caused by other (natural) effects than traffic. The rainy weather in June 2020
(the so-called St. Medard's forty days of rain in Central European folklore) yielded very similar
low pollution levels.

The study revealed that intentional reduction of traffic intensity can have unambiguous
potentials in improving urban air quality as far as NO, NO$_2$, CO and particle number
concentrations are concerned. It should be added that all smog alerts in Budapest were
exclusively announced because of PM$_{10}$ mass, which did not seem to be considerably affected
by vehicle flows. Nevertheless, measures for tranquillizing urban traffic can contribute to
improved air quality through a new strategy for lowering the population exposure of inhabitants
instead of high-risk management of individuals.

The method could be expanded by other important chemical species such as soot and by other
location types such as near-city or regional background sites jointly with the centre in order to
obtain more exact meteorology-normalized changes. The results also point to the importance
of non-linear relationships among precursors and secondary pollutants, which are to be studied
more intensively. Finally, it should be mentioned that contemporary urban air quality and
climate issues and their related policies are largely biased by financing possibilities and
economic performance/growth.

*Data availability.* The observational data are accessible at http://www.levegominoseg.hu/ or are
available from the corresponding author – except for the vehicle road traffic – upon reasonable request.
*Supplement.* The supplement related to this article is available online.
*Author contributions.* IS conceived the study. AZGy, WT and IS performed most aerosol and
meteorological measurements. All co-authors participated in the data processing and interpreting the
results. The figures were created by MV and AZGy. IS wrote the manuscript with comments from all
coauthors.




*Competing interests.* The authors declare that they have no conflict of interest.


*Acknowledgements.* This research was supported by the Hungarian Research, Development and Innovation Office (grant nos. K116788 and K132254). The authors thank the leaders of the Budapest Public Roads Ltd. (Budapest Közút Zrt.) for providing the vehicle road traffic data and its coworker Dezső Huszár for valuable discussions. The map in Fig. 1 was created by Márton Pál, Ph. D. student of the Department of Cartography and Geoinformatics, Eötvös University.

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
