# Peer review of "What can we learn about urban air quality"

_Atmospheric Chemistry and Physics, 2020_

## Referee Comment (RC1) · Anonymous Referee #1 · 12 Oct 2020

General remarks

This manuscript investigated the impact of the partial corona-virus lock-down on the air quality in Budapest. This topic, albeit a case study, has a high international value due the high general interest in such kinds of studies and because of the transferability and generalizability of these results. The specific impact of the reduced traffic intensity is studies with the help of long-term measurements, which allowed for a comparison of the 2020 data with several previous years. The methods are well described, the results are instructive, and the conclusions are generally drawn correctly. I have only a few comments that may be addressed in a revised version.

[Figure]

Minor comments

L29. I am not sure what is meant here. Did you intend to refer to "the possible role of atmospheric vehicle-induced mixing processes"?

L208: This unit conversion is based on the Magnus-equation.

L210: The coefficients of this Magnus-equation can only be applied above liquid water. Different coefficients A and B are needed above ice surfaces, i.e. T < 0 °C.

L278: Before, in the methods section you stated that you report absolute humidity and you report values for relative humidity.

Figure 7: Could you please show this plot also for vehicle circulations, which would help to explain the observed differences in air pollutant concentrations.

Figure 8 and 9: Please state that these plots show reanalysis data in the figure caption, so that the figure can be understood by itself.

L834: Is the expression in brackets really necessary?

L855-L857: It is not clear to me what is meant by this sentence and how this conclusion is supported by the presented results.
* * *

---

## Short Comment (SC1) · 12 Oct 2020

The article is very different from the normal flow of COVID-19 ENVIRONMENTAL impact research. But I am looking reference and citation, the citation is missing but the reference is there. Please check this reference Lal, P., Kumar, A., Kumar, S., Kumari, S., Saikia, P., Dayanandan, A., Adhikari, D., and Kh.......................Citation is missing.
* * *

---

## Referee Comment (RC2) · Anonymous Referee #2 · 24 Nov 2020

Salma et al. investigated the temporal variations of gaseous and particulate pollutants, including NO, $NO_2$, CO, $O_3$, $SO_2$, $PM_{10}$, $PM_{2.5}$, total particle number ($N_{6-1000}$) and particles with a diameter less than 100nm from 1 January to 31 July in 2017, 2018, 2019 and 2020 in central Budapest, Hungary. Based on the relative difference and standardized anomaly methods, the authors re-evaluated the relationships between urban air quality and motor vehicle road traffic. The authors found that the concentrations of critical particulate pollutants did not seem to be largely affected by vehicles in Budapest during the COVID-19 pandemic. The study highlighted the synergetic effect of local meteorological conditions, long-range transport and anthropogenic emissions on air quality, and also showed importance of non-linear relationships among precursors and secondary pollutions in Budapest, Hungary. I would like to recommend acceptance of the manuscript after the following comments are addressed.

1. Line 261-269: in this paragraph, the authors stated that the concentration of chemical species was based on the reanalyzed results of seven state-of-the-art European models. Please provide more descriptions about these models. If the reanalyzed data are publicly accessible, please provide a statement on how the data can be accessed.

2. In section 3.1, the alterations in the T, RH, AH, WS, GRad and $PBLH_{max}$ in the average reference year and year 2020 during the COVID-19 pandemic are quantified separately. How about the changes of wind direction? Previous studies indicated that a structure of convergence and divergence from the surface to the middle level of the troposphere also plays an important role in air pollution, so how about the convergence and divergence in the vertical direction over Budapest during the COVID-19 pandemic? The following paper is recommended for the discussion: Wu, J., Bei, N., Hu, B., Liu, S., Zhou, M., Wang, Q., Li, X., Liu, L., Feng, T., Liu, Z., Wang, Y., Cao, J., Tie, X., Wang, J., Molina, L. T., and Li, G.: Aerosol–radiation feedback deteriorates the wintertime haze in the North China Plain, Atmos. Chem. Phys., 19, 8703–8719, https://doi.org/10.5194/acp-19-8703-2019, 2019.

3. In Table S2, the median of hourly mean GRad in 2020 is less than that in the average reference year, and the lower radiation could suppress the development of the PBL, but the $PBLH_{max}$ in 2020 is higher than that in the average reference year. Please provide explanation for this phenomenon.

4. Line 277-280: Please provide quantitative results or references to explain why Spring 2020 is the third driest season since 1901.

5. Figure 8 and 9: In the figure caption, please clarify whether the data used are observations or simulations.

---

## Author Comment (AC1) · 8 Dec 2020

**Response to Referee number 1**

8th December 2020

The authors would like to thank Referee no. 1 very much for his/her expertise and valuable comments to further improve and clarify the MS. We also appreciate his/her quick reaction. We have considered all recommendations and made the appropriate alterations. We also accomplished some other smaller corrections. Our specific responses are as follows, while the textual modifications amended can be traced in the marked-up version of the MS, which is attached.

**Minor comments**

L29. I am not sure what is meant here. Did you intend to refer to "the possible role of atmospheric vehicle-induced mixing processes"?

1. The sentence in Abstract was shortened by removing its unclear part.

L208: This unit conversion is based on the Magnus-equation.

2. The description of the conversion method was clarified by several, smaller textual modifications. See also Answer no. 3.

L210: The coefficients of this Magnus-equation can only be applied above liquid water. Different coefficients A and B are needed above ice surfaces, i.e. T < 0 _C.

3. Equation 1 with identical coefficients is acceptable for sub-cooled liquid water as well. We explicitly indicated in the text now that we utilized this approximation. The lowest 1-h mean *T* was ca. –5 °C, which confirms that this seems to be a plausible approach since the two saturation vapour pressure curves for liquid water and ice surface follow each other closely in the related temperature range. Moreover, the freezing occurred from January to the beginning of April, and, therefore, the conversion did not affect the restriction phases which were in the focus of the study.

L278: Before, in the methods section you stated that you report absolute humidity and you report values for relative humidity.

4. We performed the conversion of RH to AH in order to facilitate future comparison with other locations or cities in the world mainly for possible virology purposes. This conversion

requires the *T* data, and, therefore, the reader cannot accomplish it. For the objectives of the present study, however, the RH seems to be more relevant than AH and, therefore, we kept discussing the original property. The sentence explaining the purpose for the conversion was amended accordingly.

Figure 7: Could you please show this plot also for vehicle circulations, which would help to explain the observed differences in air pollutant concentrations.

5. The diurnal variations of vehicle circulation for all pandemic phases were shown Fig. S5 in the Supplement. On the request of the Referee, we extended now Fig. 7 by the panel showing the relevant vehicle circulation curves in the restriction phase, and added some further explanations. In addition, we emphasized better now in a proper place in the text that the corresponding plots can be find in the Supplement.

Figure 8 and 9: Please state that these plots show reanalysis data in the figure caption, so that the figure can be understood by itself.

6. The required extension was added to the figure captions.

L834: Is the expression in brackets really necessary?

7. The expression in brackets was deleted.

L855-L857: It is not clear to me what is meant by this sentence and how this conclusion is supported by the presented results.

8. The sentence indicated was removed from the MS.

Imre Salma
corresponding author

---

## Author Comment (AC2) · 8 Dec 2020

**Response to Referee number 2**

8th December 2020

The authors would like to thank Referee no. 2 very much for his/her expertise and valuable comments to further improve and clarify the MS. We have considered all recommendations and made the appropriate alterations. We also accomplished some other smaller corrections. Our specific responses are as follows, while the textual modifications amended can be followed in the marked-up version of the MS, which is attached.

1. Line 261-269: in this paragraph, the authors stated that the concentration of chemical species was based on the reanalyzed results of seven state-of-the-art European models. Please provide more descriptions about these models. If the reanalyzed data are publicly accessible, please provide a statement on how the data can be accessed.

Further details of the CAMS modelling method utilized were summarized and provided with a reference for the on-line availability of the model.

2. In section 3.1, the alterations in the T, RH, AH, WS, GRad and PBLHmax in the average reference year and year 2020 during the COVID-19 pandemic are quantified separately. How about the changes of wind direction? Previous studies indicated that a structure of convergence and divergence from the surface to the middle level of the troposphere also plays an important role in air pollution, so how about the convergence and divergence in the vertical direction over Budapest during the COVID-19 pandemic? The following paper is recommended for the discussion: Wu, J., Bei, N., Hu, B., Liu, S., Zhou, M., Wang, Q., Li, X., Liu, L., Feng, T., Liu, Z., Wang, Y., Cao, J., Tie, X., Wang, J., Molina, L. T., and Li, G.: Aerosol–radiation feedback deteriorates the wintertime haze in the North China Plain, Atmos. Chem. Phys., 19, 8703–8719, https://doi.org/10.5194/acp-19-8703-2019, 2019.

We demonstrated earlier that the local wind direction (WD) at the BpART Laboratory is strongly modified by the built urban environment and local orography with respect to the synoptic wind (Salma et al.: Measurement, growth types and shrinkage of newly formed aerosol particles at an urban research platform, Atmos. Chem. Phys., 16, 7837–7851, 2016, Sect. 3.2 and Fig. 5). This is the reason why we did not investigate directly the variations in WD. The long-range transport of air masses was, however, involved in the study through the macrocirculation patterns determined specifically for the geographical area. As far as the convergence and divergence in the vertical direction (as we understand, in the change of the vertical wind velocity) over Budapest is concerned, it does not seem to be substantially influence the air quality in the city

because of limitations constrained by the actual geographical location. We are aware that the vertical wind distribution could be connected e.g. to the heat island intensity, which is implicitly contained in the PBLH, and this latter quantity was indeed involved in the evaluations. The feedback mechanism discussed in the mentioned article is not relevant for us because 1) it occurs outside the COVID-19 time interval, and 2) the poor air quality in Budapest is usually associated with long-term cold air pool above the Carpathian Basin in winter, but it is accompanied by foggy situations and low radiation instead of wintertime haze, which is typical for the North China Plain. We would like to thank you very much for this comment because it triggered us to add a new section 3.8 Potentials as a follow up of this remark, where we could explain in more detail all this and the role of the $T$ inversions for air quality issues in the Carpathian Basin in wintertime.

3. In Table S2, the median of hourly mean GRad in 2020 is less than that in the average reference year, and the lower radiation could suppress the development of the PBL, but the PBLHmax in 2020 is higher than that in the average reference year. Please provide explanation for this phenomenon.

The median GRad data (for $\geq 50$ W m$^{-2}$) was lower in Y2020 by ca. 2.5 % than in Y3Ref, while the median PBLH$_{max}$ value was larger in Y2020 by 10 % than in Y3Ref. We think that the two differences are insignificant when comparing them to the uncertainty intervals (in particular, for the modelled PBLH$_{max}$) and when considering the effects of some other confounding meteorological variables such as precipitation. We would not draw any conclusion on the relationships of GRad and PBLH$_{max}$ based on such small differences. A short sentence dealing with this was added to the text to avoid any misunderstanding.

4. Line 277-280: Please provide quantitative results or references to explain why Spring 2020 is the third driest season since 1901.

The extremely dry spring in 2020 can likely be related to multifactorial meteorological reasons. Between 14 March and 24 April, anti-cyclonic weather types prevailed in the Carpathian Basin almost continuously for 41 days (Table S2). After this interval, the weather type was mostly cyclonic but with northerly wind, which brings dry and could air masses into the Budapest area. These factors together resulted in the drought experienced. This was also briefly added to the MS.

5. Figure 8 and 9: In the figure caption, please clarify whether the data used are observations or simulations.

The captions of the figures were extended to include the requested information.

Imre Salma
corresponding author

---

## Author Comment (AC3) · 9 Dec 2020

Dear Colleague,

we would like to thank you for this comment, which was adopted into the new MS.

Imre Salma corresponding author